# A Sequential Membrane Process of Ultrafiltration Forward Osmosis and Reverse Osmosis for Poultry Slaughterhouse Wastewater Treatment and Reuse

**DOI:** 10.3390/membranes13030296

**Published:** 2023-03-01

**Authors:** Faryal Fatima, Hongbo Du, Raghava R. Kommalapati

**Affiliations:** 1Center for Energy and Environmental Sustainability, Prairie View A&M University, Prairie View, TX 77446, USA; 2Department of Civil and Environmental Engineering, Prairie View A&M University, Prairie View, TX 77446, USA

**Keywords:** poultry slaughterhouse wastewater, ultrafiltration, reverse osmosis, chemical oxygen demand, total phosphorous, total solids, total volatile solids, total fixed solids, total nitrogen

## Abstract

To address some challenges of food security and sustainability of the poultry processing industry, a sequential membrane process of ultrafiltration (UF), forward osmosis (FO), and reverse osmosis (RO) is proposed to treat semi-processed poultry slaughterhouse wastewater (PSWW) and water recovery. The pretreatment of PSWW with UF removed 36.7% of chemical oxygen demand (COD), 38.9% of total phosphorous (TP), 24.7% of total solids (TS), 14.5% of total volatile solids (TVS), 27.3% of total fixed solids (TFS), and 12.1% of total nitrogen (TN). Then, the PSWW was treated with FO membrane in FO mode, pressure retarded osmosis (PRO) mode, and _L_-DOPA coated membrane in the PRO mode. The FO mode was optimal for PSWW treatment by achieving the highest average flux of 10.4 ± 0.2 L/m^2^-h and the highest pollutant removal efficiency; 100% of COD, 100% of TP, 90.5% of TS, 85.3% of TVS, 92.1% of TFS, and 37.2% of TN. The performance of the FO membrane was entirely restored by flushing the membrane with 0.1% sodium dodecyl sulfate solution. RO significantly removed COD, TS, TVS, TFS, and TP. However, TN was reduced by only 62% because of the high ammonia concentration present in the draw solution. Overall, the sequential membrane process (UF-FO-RO) showed excellent performance by providing high rejection efficiency for pollutant removal and water recovery.

## 1. Introduction

The consumption of meat has greatly increased along with the rapid growth in the human population. Compared to other meats such as beef or pork, the daily intake of poultry (particularly chicken) is increasing much more rapidly [1]. In 1996, the consumption rate of poultry meat was 9 million tonnes per year (Mt/year) which increased to 133 Mt/year in 2020 [2]. The poultry industry uses an average of 15–20 L/bird of fresh water, and 80 to 90% of this process water is discharged as poultry slaughterhouse wastewater (PSWW) [3]. Freshwater resources and municipal wastewater treatment facilities are under significant strain due to the rising demand for poultry meat in the United States and worldwide. In addition, the conventional PSWW methods and even advanced oxidation processes that are physiochemical and biological processes do not purify the water to the quality level of potable water. Therefore, reusing wastewater is an excellent approach to the sustainability of water resources [4].

Pressure-driven membrane technology is one of the solutions for the treatment and reuse of PSWW [5]. The four types of pressure-driven membranes are microfiltration (MF), ultrafiltration (UF), nanofiltration (NF), and reverse osmosis (RO). RO membrane has been widely used for water reclamation because of its ability to separate tiny particles and monovalent ions. For example, a RO membrane can remove sodium and chloride ions with up to 99.5% efficiency. Moreover, the MF, UF, and NF usually serve as pretreatment steps for RO. The pressure-drive membrane has made water recovery from wastewater a suitable option. However, the energy requirement remains a significant challenge [6].

In addition to the pressure-driven membrane, the forward osmosis (FO) membrane is also used for wastewater treatment. FO is an advanced technology that provides several advantages over pressure-driven membrane processes, such as lower energy input, decreased fouling tendency, easier fouling removal, and high-water recovery [7]. However, the main disadvantage of FO is that the product water of FO is a diluted draw solution, so a post-treatment, e.g., RO, is required to produce clean water. Another challenge that FO faces is that it is difficult to find a suitable draw solute that can generate high osmotic pressure and is simple to recover or regenerate with a very low cost [8]. The FO membrane fouling is lesser than pressure-driven membranes, but it still affects membrane performance. One commonly used antifouling method for FO membranes is surface modification [7]. The surface modification also improves the water flux by lowering the membrane fouling. The surface modification process can be carried out in various ways, such as physical adsorption, surface coating, and chemical vapor deposition [9]. In recent times zwitterion materials have been studied for their antifouling properties because of their high electroneutrality and hydration capacity. Zwitterion materials have a strong ability to resist bacterial adhesion and biofilm development [10]; _L_-DOPA is one such zwitterionic polymer that has been used to enhance membrane surface antifouling capabilities.

Compared to conventional wastewater treatment methods, the FO process has several advantages and exhibits promising outcomes for wastewater treatment. Its high performance in water recovery without highly-driven pressure enables the viability of FO processes. Additionally, FO offers a more sustainable flux and pollutant removal. Since the 18th century, over 1000 FO publications have been reported for FO membranes and their applications. The research on FO has been mainly on municipal wastewater, oily wastewater, tanner wastewater, automobile effluents, dairy streams, and produced wastewater. However, to our knowledge, no research has been conducted on PSWW using FO [11,12]. This study explores the performance of FO for the first time for the treatment and reusing PSWW treatment. It will also provide ideas to other scientists for future research on FO membrane challenges for meat slaughterhouse wastewater treatment and reuse.

The sequential membrane process (UF-FO-RO) included in the entire process of PSWW treatment, as shown in Figure 1, will address these challenges of food security and sustainability of the poultry processing industry by removing all the contaminants from wastewater and recycling water in the poultry industry. The proposed sequential membrane processes (UF-FO-RO) will improve food safety and sustainability during poultry processing. It will also resolve ecological and environmental problems, such as depleting freshwater resources, spreading foodborne contaminants via inefficiently treated wastewater, rising operating costs of poultry processing plants, and growing nutrient pollution in watersheds.

## 2. Materials and Methods

### 2.1. Characterization of Poultry Slaughterhouse Wastewater

PSWW samples were collected from Sanderson Farms, located in Bryan, Texas. Sanderson Farms is one of the largest poultry producers in the United States, and the company owns its PSWW treatment plant. The samples were collected in a plastic container of 5 gallons from the discharging stage further to purify it for the recycling limits of wastewater. For each 5-gallon container, 10 lbs. of ice were used during the delivery of PSWW from Sanderson Farm to the research lab to avoid degradation of PSWW. The samples at the lab were stored in a refrigerator at 4 °C. All parameters of wastewater were measured for turbidity, chemical oxygen demand (COD), total solids (TS) total dissolved solids (TDS), total suspended solids (TSS), total volatile solids (TVS), total fixed solids (TFS), total nitrogen (TN) and total phosphorous (TP) according to the Standard Methods for the Examination of Water and Wastewater, as listed in Table 1. Hach instruments, reagents, and vials were used for COD, TN, and TP examination. The instruments, reagents and vials were purchased from Hach Company (Loveland, Colorado, USA). An average of three trials were considered for all the characterization tests. All the instruments involved in measuring COD, TN, and TP were tested for accuracy using their given standards.

### 2.2. Treatment of PSWW with Ultrafiltration

The PSWW was purified with a pressure-driven membrane filtration system purchased from Sterlitech Corporation (Auburn, Washington, DC, USA). The membrane material is polyethersulfone (PES) with a molecular weight cut-off (MWCO) of 30,000 Da, as listed in Table 2. The membrane system consisted of an acetal (Delrin) cell with an outer dimension of 0.127 m length, 0.1 m width, 0.083 m height, and an effective membrane area of 0.0042 m^2^. The UF membrane filtration was operated at a pressure of 827 kPa and a flow rate of 6.0 L/minute. The dynamics of the membrane flux were monitored, and when it was reduced to below 50% after 2 h, the operation was stopped, and the membrane was washed to restore flux. To restore the flux, the membrane was flushed with DI water for 45 min. Then the membrane was rinsed with 0.1% sodium hydroxide solution and a light bleach concentration for 45 min at 48.8 °C. Subsequently, the membrane was washed with 0.2% phosphoric acid for 45 min at room temperature. Afterward, the membrane was flushed with DI water for 45 min, and then the system was again operated to purify PSWW.

The performance of the UF process was examined by evaluating permeate flux and characterizing the permeate quality. Based on the data gathered, average flux (Jav), rejection coefficient (*R*), volume concentration ratio (*VCR*), and real-time flux (Jw) were determined. The *R* and *VCR* were calculated with Equations (1) and (2).
(1)$$R=1-\frac{concentration\;of\;permeate}{concentration\;of\;retenate}$$
(2)$$VCR=\frac{initial\;feed\;volume}{retentate\;volume}$$

### 2.3. Preparation of Forward Osmosis with Surface Modification

The FO membrane was coated with 3-(3,4-dihydroxy phenyl)-L-alanine (_L_-DOPA). _L_-DOPA is a zwitterion (redox functional amino acid) that self-polymerizes in aqueous solutions and forms a strong bond with various substrates. The _L_-DOPA coating improved membrane hydrophilicity and fouling resistance considerably. For coating the FO membrane, an _L_-DOPA was obtained from Sigma-Aldrich (USA); it was dissolved entirely in the 10 mM Tris-HCl pH 8.0 buffer solution using a magnetic stirrer. A 10 mM Tris HCl pH 8.0 buffer solution was prepared by dissolving 12.1 g of Tris base in 80 mL of nuclease-free water and then by adding 6 mL concentrated HCl to give a 1 M Tris-HCl buffer solution with a pH of 8.0. After that, 1 M Tris-HCl was diluted with 1:100 to obtain a 10 mM Tris- HCl pH-8.0 solution. The CF042 FO system purchased from Sterlitech Corporation was used to coat the support layer of the cellulose triacetate (CTA) FO membrane by circulating the coating solution. The specification of the original FO membrane is listed in Table 3. For coating the membrane, 1000 mL of 2 g/L _L_-DOPA solution was used on the feed side, and 1000 mL of DI water was used on the draw side at a flow rate of 3.0 L/min. The membrane was coated on the porous side for the optimized time of 12 h [14]. After coating the membrane, it was washed with DI water for 15 min and stored at 4 °C in a refrigerator. The membrane was characterized by measuring surface hydrophilicity and Fourier-transform infrared (FTIR) spectroscopy.

Water contact angle analysis was used to determine the membrane surface’s hydrophilicity. A CAM-PLUS contact angle meter was purchased from ChemInstruments (West Chester Township, OH, USA), and it works on the half-angle measuring principle. FTIR spectroscopy was used to determine the functional groups present in the virgin and coated membrane using a Smiths Detection spectrometer (Lakeside Boulevard, MD, USA).

### 2.4. Treatment of PSWW with Hybrid FO-RO Process

For this research, a lab-scale FO system was used, and it consisted of an acrylic membrane cell with an outer dimension of 0.127 m in length, 0.1 m in width, 0.083 m in height, and an effective membrane area of 0.0042 m^2^. It also contained two gear pumps, two beakers containing feed and draw solution, a balance, and a computer, as shown in Figure 2. The CTA membrane, as listed in Table 3, was purchased from Sterlitech for the FO system. The FO system was first evaluated at three different flow rates of 3.0, 2.5, and 2.0 L/min using 1000 mL of DI water as a feed solution and 1000 mL of 3 M ammonia–carbon dioxide as a draw solution. Then the PSWW was purified using 1000 mL of UF permeate as a feed solution and 1000 mL of 3 M ammonia–carbon dioxide as a draw solution at an optimized flow rate. The membrane was cleaned every 7 h to restore the membrane flux. First, the membrane was flushed with DI water for 30 min. Then 0.1% sodium dodecyl sulfate was recirculated through the FO system for 45 min at room temperature. Lastly, the membrane was rinsed again with DI water for 30 min.

The membrane performance was investigated by measuring the permeate flux and characterizing the permeate quality. We ran the FO process with virgin and coated membranes in two modes of FO and pressure-retarded osmosis (PRO). Before characterization, the FO permeate was heated at 68 °C to get rid of ammonia gas. The Jav, VCR, and Jw were calculated using the same approach as used to analyze UF membrane performance. For FO permeate, reverse solute flux was determined by taking the difference between the initial and final feed concentration multiplied by the initial and final volume and then dividing by the area of the membrane and time interval, which was expressed in g/m^2^-h, as shown in Equation (3),
(3)$$Reverse\;Solute\;Flux=\frac{C_{t}V_{t}-C_{0}V_{0}}{A*\Delta t}$$
where *C*_0_ is the original concentration, *V*_0_ is the original volume, *C_t_* is the concentration at time *t*, *V_t_* is the volume at time *t*, *A* is the effective membrane area, and Δ*t* is the running time starting from the original time point.

The FO permeate was separated from the draw solution using the RO process. The same Sterlitech membrane filtration system was used for RO with proper pressure setup. The RO was run in the crossflow mode at the 6.0-L/min flow rate and 2895 kPa pressure. The specification of the membrane is listed in Table 4. The membrane performance was investigated by characterizing the permeate quality.

### 2.5. Investigation of Membrane Fouling

The fouling of the UF and FO membrane was investigated based on reversibility and irreversibility. The foulants of the UF membrane were analyzed by a Smith Detection FTIR spectrometer and PerkinElmer thermogravimetric analyzer (Austin, TX, USA). For the FTIR spectroscopy, the foulants were collected from the membrane surface, air-dried at room temperature, and then tested for detecting functional groups. The weight percentage of moisture, organic and inorganic compounds present in the foulants was determined by thermogravimetric analysis (TGA). The TGA was conducted starting from room temperature, increasing by 10 °C /min and reaching up to 800 °C.

## 3. Results

### 3.1. Poultry Slaughterhouse Wastewater Characteristics

The characterization of PSWW was conducted with samples collected at the discharging stage of the Sanderson Farms, after UF treatment, after FO treatment, and after RO treatment. Most of the pollutants present in PSWW were various salts; therefore, the TS was high, and the COD was low. The TS was 1830 mg/L, and the COD level was 30 mg/L, as reported in Table 5. The TVS and TFS were measured as 366 mg/L, and 1464 mg/L, respectively. Moreover, no TSS was detected in PSWW; hence, all the solids were dissolved solids. The concentration of TP was 36 mg/L, and the level of TN was 107 mg/L. The value of TN is very high because the conventional wastewater treatment at the Sanderson plant does not remove all the nitrogen from PSWW. Some of the parameters of PSWW qualified the requirement for discharging it to the environment; however, they did not meet the requirement for recycling limits of wastewater.

### 3.2. Pretreatment of PSWW with Ultrafiltration

#### 3.2.1. UF Permeate Characterization

The PES UF membrane was used for the pretreatment of PSWW. The UF process somewhat reduced the pollutant levels of PSWW since a small quantity of macromolecules were present in the PSWW at the discharging step. The UF membrane reduced 36.7% of COD, 12.1% of TN, and 38.9% of TP. In addition, the removal efficiency of 24.7% of TS was achieved, along with a reduction of 14.5% of TVS and 27.3% of TFS. The UF permeate was slightly clear in color as compared to raw PSWW.

Previous research on untreated PSWW effluent has reported high removal efficiency for the UF membrane, as shown in Table 6. For example, Jason et al. [18] treated raw PSWW with UF membrane. The results showed that the UF significantly reduced 85% of TS, 95% of COD, and 86% of TN for raw PSWW. On the other hand, this project did not achieve high removal efficiency for the PES UF membrane because the PSWW collected already went through conventional wastewater treatment, which removed most organic compounds.

#### 3.2.2. Real-Time Flux & Average Flux of UF Permeate

The pretreatment of PSWW with UF membrane was conducted in three trials. Each trial was run for 2 h, and then the membrane was cleaned to restore the flux. The results showed that the cleaning method of the UF membrane was efficient for restoring the flux as the 0.1% sodium hydroxide solution removed almost all the fouling for organic matter and 0.2% phosphoric acid solution released fouling for the inorganic material. The highest permeate flux achieved for the virgin membrane was 116.5 ± 0.15 L/m^2^-h, as shown in Figure 3. The flux of the membrane after the 1st wash and 2nd wash was successfully restored to 94% (109.2 ± 1.05 L/m^2^-h), and 87% (101.6 ± 1.1 L/m^2^-h), respectively. The average flux achieved was 78.5 ± 0.5 L/m^2^-h for the virgin membrane, 63.5 ± 4.6 L/m^2^-h after 1st wash, and 59.7 ± 2.3 L/m^2^-h after 2nd wash. Coskun et al. [20] treated PSWW treatment using laboratory-scale membrane processes. At 200 kPa, the bench-scale UF membrane had the highest permeate flux of 112.1 L/m^2^-h, which is close to the flux of 116.3 L/m^2^-h achieved in this project.

#### 3.2.3. Volume Concentration Ratio & Rejection Coefficient of UF Permeate

For the pretreatment of PSWW with UF membrane, an initial feed of 12 L PSWW was used; at the end of the process, 3 L was left as retentate; therefore, the VCR was 4. In addition, for the UF permeate, the concentration of TDS of feed was 1580 ppm, and the retentate TDS concentration was 1650 ppm; therefore, the rejection coefficient was 0.4. The 0.4 R is justifiable by seeing Table 6 that the reduction efficiency of TS for the UF process was low. It is because there are larger pores in UF membranes compared to NF and RO membranes.

### 3.3. Preparation of Forward Osmosis with Surface Modification

The 12-h optimizing time recommended by Kommalapati et al. [14] for coating the membrane with _L_-DOPA showed excellent results for this project. The _L_-DOPA was well deposited on the porous side of the membrane, as shown in Figure 4.

Membrane surface hydrophilicity has a significant impact on membrane fouling resistance. A contact angle analysis was conducted to investigate the hydrophilicity of membrane surfaces. It was observed that the wettability of the coated membrane was higher than the virgin membrane as the contact angle was reduced from 46.3° to 36.5°. This is due to positively and negatively charged _L_-DOPA polymers coated on the membrane surface, which interact strongly with water via an ionic–dipole interaction. These two charges considerably contribute to the significant hydrophilicity of _L_-DOPA molecules.

The FTIR spectra of the virgin FO membrane and the coated membrane are shown in Figure 5. The functional groups present in the CTA FO membrane and _L_-DOPA coated membrane are similar. The FTIR spectra mainly detected three functional groups: -C-O-C-, -C=O, and -O-H. The characteristic peak of OH groups appears at roughly 3400 cm^−1^. The -C=O group was attributed to the band occurring at approximately 1700 cm^−1^, and the -C-O-C- bond was assigned to the peak at around 1200 cm^−1^, as shown in Figure 5. Compared to the uncoated membrane, the band intensity of the O-H group was decreased for the coated membrane. This is primarily because of the chemical interaction between the catechol group (1,2-dihydroxybenzene) in the _L_-DOPA and the -O-H group on the untreated CTA membrane.

Azari et al. [7] concluded that direct coating of zwitterionic _L_-DOPA on the porous side of commercial CTA FO membrane reduced the contact angle from 48° to 38° and that the coated membrane surface became more hydrophilic. Their FTIR examination detected similar functional groups, including -C-O-C-, -C=O, and -O-H.

### 3.4. Treatment of PSWW with Hybrid FO-RO Process

#### 3.4.1. Flowrate Optimization of the Circulating Flow Rates

For choosing the optimal flow rate, the CTA FO membrane was run with three different flow rates, 3.0, 2.5, and 2.0 L/min using 1000 mL of DI water as a feed and 1000 mL of 3 M ammonia–carbon dioxide draw solution. Each flow rate was tested for almost 7 h. At the end of the 7-h testing period, average flux and reverse solute flux were calculated for each flowrate. The average and reverse solute flux at all three flow rates were almost equal. The real-time flux pattern was also similar, as shown in Figure 6. The average flux and reverse solute flux for 3.0, 2.5, and 2.0 L/minute flow rates were 11.8 L/m^2^-h and 50.4 g/m^2^-h, 11.7 L/m^2^-h and 50.6 g/m^2^-h, 11.6 L/m^2^-h and 51.6 g/m^2^-h, respectively. The 3.0 L/minute was the optimal flow rate since the average water flux was the highest and the reverse solute flux was the lowest. We need to point out that the water flux does not always increase with the increase of the crossflow rate of feeding water because a high crossflow rate can cause some issues, e.g., feed channel pressure drop [21].

#### 3.4.2. Permeate Characterization of FO Mode, PRO Mode, and Coated Membrane

The characterization of FO permeate was conducted for the FO and PRO mode and coated membrane in PRO mode. The FO mode showed excellent performance in terms of the rejection of various pollutants; it achieved the highest removal efficiency for all the parameters of wastewater characterization. The FO mode produced permeate with 100% removal of COD and TP. It removed the TS from 1830 to 130 mg/L, as shown in Table 7, thus achieving a removal efficiency of 90.5% of TS. The reduction percentage for TVS and TFS was approximately 85.3% and 92.1%, respectively. TN was reduced from 94 to 59 mg/L.

The PRO and coated membrane also achieved complete retention of COD and TP. However, the removal efficiency of PRO mode for other pollutants was higher than the coated membrane. PRO mode removed 84.2% of TS, 83.3% of TVS, 87.6% of TFS, and 12.8% of TN. Additionally, the coated membrane in the PRO mode removed 79% of TS, 80.6% of TVS, 75.9% of TFS, and 6.3% of TN in the concentrated retentate.

The main reason for the FO mode achieving high removal efficiency compared to the PRO mode is that in FO mode, the active layer, which is very smooth with no pores, faces the feed solution, and it is very effective in retaining the pollutants in the feeding side. However, in the PRO mode, some pores in the support layer was blocked by the pollutants after running the system for some time, thus decreasing the membrane’s overall performance. Moreover, the PRO mode in coated membrane showed the least removal efficiency because the inorganic particles in the feed were incompatible with the zwitterionic polymer to remove them.

Ezugbe et al. [22] researched the desalination of municipal wastewater using FO. They used a similar CTA FO membrane operated in the FO mode to desalinate the wastewater. Their results showed that the FO membrane reduced COD by approximately 60% [22]. However, the efficiency achieved in this project for COD removal using the same FO membrane was 100%, mainly because the PSWW was pretreated with UF. Gao et al. [23] examined an anaerobic FO membrane bioreactor for treating municipal sewage. The study reported that the FO membrane showed high removal efficiency of 96%, 89%, and 100% of COD, TN, and TP, respectively, as shown in Table 8, confirming that pretreatment is essential to increase the removal efficiency of the FO process.

Goa et al. [24] directly treated municipal sewage by FO and investigated membrane fouling behaviors; the results showed that the removal efficiency of FO was 96.5%, 89.4%, and 95.4% of COD, TN, and TP, respectively, but severe membrane fouling occurred. However, the removal efficiency of TN achieved in this project was 37.8% of TN because the draw solution used was ammonia-carbon dioxide. The fouling was negligible for this project because the PSWW removed most organic and inorganic pollutants after UF pretreatment.

#### 3.4.3. Real-Time Flux & Average Flux in FO Mode, PRO Mode, and Coated Membrane

The treatment of PSWW with FO membrane was conducted in three trials. Each trial was run for almost 7 h, and then the membrane was cleaned to restore the flux. The flux in FO mode, PRO mode, and the coated membrane in PRO mode was restored to almost 100% by flushing the membrane with 0.1% sodium dodecyl sulfate solution. In addition, the pattern of the real-time flux for the virgin and washed membranes were very similar to one another for FO, PRO, and coated membranes, as shown in Figure 7, Figure 8 and Figure 9. The membrane performance was restored to 100% by completely removing the fouling.

The virgin and washed membrane initially achieved the highest flux in the FO mode, almost 10 L/m^2^-h, slowly decreasing to approximately 8 L/m^2^-h in 7 h. The average flux in the FO mode achieved for the virgin membrane was 10.4 ± 0.2 L/m^2^-h. The average flux was restored to 9.9 ± 0.5 L/m^2^-h, and 9.8 ± 0.15 L/m^2^-h after 1st wash and 2nd wash, respectively. The virgin and washed membrane achieved the highest flux for the PRO mode, almost 9 L/m^2^-h at the beginning, which declined to 5–6 L/m^2^-h in 1.5 h and slowly decreased to about 4 L/m^2^-h in 7 h. The average flux of the PRO mode was around 6.6 ± 0.5 L/m^2^-h for the virgin membrane, 5.9 ± 0.7 L/m^2^-h for 1st wash, and 5.5 ± 0.1 L/m^2^-h for 2nd wash. For the coated membrane in PRO mode, the virgin membrane and the washed membrane achieved the highest flux, almost 17 L/m^2^-h, which slowly decreased to approximately 3 L/m^2^-h in 7 h. The average flux of the coated virgin membrane was about 6.3 ± 0.2 L/m^2^-h, which was restored to 6.2 ± 0.6 L/m^2^-h after 1st wash, and 6.1 ± 0.1 L/m^2^-h after 2nd wash.

Zhang et al. [25] researched municipal processing wastewater by FO using a CTA membrane. They achieved 7.4 L/m^2^-h flux for FO mode. Their flux is less than the flux achieved in this project because their water was raw municipal wastewater. The PSWW used in this project was pretreated with conventional wastewater treatment methods, and then it was further treated with the UF process.

Moreover, in the PRO mode, the flux decreased at a very high rate compared to the FO mode because the pores of the membrane support layer were eventually blocked by organic and inorganic matter present in the wastewater. In our trials, the coated membrane in PRO mode showed the least flux and membrane performance, which is opposite to our expectation of zwitterionic coating. The first reason is that the pore size of the membrane support layer is reduced by depositing _L_-DOPA. The second is that the zwitterionic coat only repels organic matter, not inorganic species, e.g., dissolved silica present in the influent. After running for some time, the inorganic matter would accumulate, depositing in smaller pores. Even further, organic matter deposits in the pores after the zwitterionic coating is covered by inorganic compounds.

#### 3.4.4. Volume Concentration ratio and Reverse Solute Flux in FO Mode, PRO Mode, and Coated Membrane

For the FO mode, the initial feed volume of UF permeate was 1000 mL, and by the end of the 7 h, the retentate volume was 770 mL; therefore, the VCR was calculated as 1.29. For VCR, the PRO mode was calculated as 1.35 as the initial feed volume was 1000 mL and the final retentate volume was 740 mL. The coated membrane in PRO mode VCR was calculated as 1.25 as the initial feed volume was 1000 mL and the final retentate volume was 800 mL. The reverse solute flux of the FO mode, the PRO mode, and the coated membrane in PRO mode was approximately 69.7 ± 5 g/m^2^-h, 66.8 ± 4.5 g/m^2^-h, and 64.3 ± 0.9 g/m^2^-h, respectively.

#### 3.4.5. PSWW Treatment with RO

A few research articles are available on the treatment of PSWW with RO; all the previous research has combined some pretreatment methods with RO to purify PSWW. The RO process separated the FO permeate from the draw solution and further removed the pollutants for this project. The RO membrane completely removed all the contaminants from the PSWW and achieved 100% removal efficiency for all the wastewater characteristics, excluding TN. The TN removal efficiency was 62.7%, as listed in Table 9, because the ammonia concentration in the draw solution was very high, and ammonia is very difficult to separate from DI water as it is extremely soluble in water.

### 3.5. Investigation of Membrane Fouling Mechanism

There are two types of fouling based on cleanability: reversible and irreversible. Reversible foulant adheres loosely to the membrane surface and can be effectively removed through physical cleaning. Irreversible foulants are tightly bound to the membrane and can only be removed through chemical cleaning [27]. When the FO membrane in FO mode, PRO mode, and the coated membrane in PRO mode were flushed with a 0.1% sodium dodecyl sulfate after PSWW treatment, it was thoroughly cleaned with no apparent fouling. The performance was restored completely, thus, confirming that the fouling was reversible. One of the main reasons is that the PSWW was collected after the discharging step and pretreated with UF; these treatments removed most of the pollutants from PSWW, which are responsible for creating a fouling layer on the membrane. The second reason is that the FO process only has much lower hydraulic pressure on micropollutant particles which can settle down on the membrane surface compared to the pressure-driven UF process. Thus, the attachment of pollutants on the membrane surface is loose and the FO membrane can thoroughly be cleaned by our washing method.

Compared to FO, the UF membrane fouling was irreversible because the flux was not restored to 100%. When the water was collected at the discharging step, the PSWW had dirt in the water and organic pollutants. Those pollutants reduced the membrane performance, and even after washing it with 0.1% sodium hydroxide and 0.2% phosphoric acid, the performance was not fully restored. The fouling was not obliterated from the membrane. The TGA results showed that the UF membrane foulants were composed of 20% water, 60% organic, and 20% inorganic compounds, as shown in Figure 10. At 200 °C all the water was evaporated, at 600 °C most of the organic compounds were degraded, and beyond 600 °C inorganic compounds remained. The FTIR spectra confirmed that the main foulants on the UF membrane were organic compounds, e.g., proteins and carbohydrates in algae and colloids. As shown in Figure 11, the peak at 1025.3 cm^−1^ presents the C-O group in polysaccharides, 1400 cm^−1^ shows the C-O group in hydroxy acid lipids, 1543.5 cm^−1^ and 1649.8 cm^−1^ indicates the amino group in protein. The 2930.5 cm^−1^ and 3297.7 cm^−1^ represent the C-H group in fatty substances and a hydroxyl group in protein, respectively.

## 4. Conclusions

In this research, the PSWW was collected from the discharging stage of the poultry slaughterhouse plant after the conventional treatment and was further purified with a sequential process of UF-FO-RO for water recycling. Our results demonstrate that UF is a promising pretreatment option for FO that can significantly reduce FO fouling and pollutant levels in PSWW. The fouling of UF was irreversible, and the main foulants of UF are protein and carbohydrates. The UF process showed removal efficiency of 36.7% of COD, 38.9% of TP, 12.1% of TN, 24.7% of TS, 14.5% of TVS, and 27.3% of TFS. The FO process is used for further purification of PSWW. Compared to the PRO mode, the FO mode was the most efficient by providing higher removal efficiency and higher average flux. The FO fouling is reversible for all operations, including FO, PRO mode, and coated membrane in PRO mode. The product water quality after RO is almost comparable to potable water except for TN. It is recommended that future studies must be conducted on the removal of TN from high nitrogen concentration draw solution for the FO process. Overall, the results show that a sequential membrane process (UF-FO-RO) is a promising approach for PSWW treatment. It exhibits excellent performance by providing high efficiency for pollutant removal and recovering valuable products. It removes almost all the pollutants and purifies the water as required to reuse for industrial poultry purposes. We expect future research to focus on the cost reduction of RO by selecting a low-energy-demand RO membrane at the end of the sequential UF-FO-RO treatment.

## Figures and Tables

**Figure 1 membranes-13-00296-f001:**
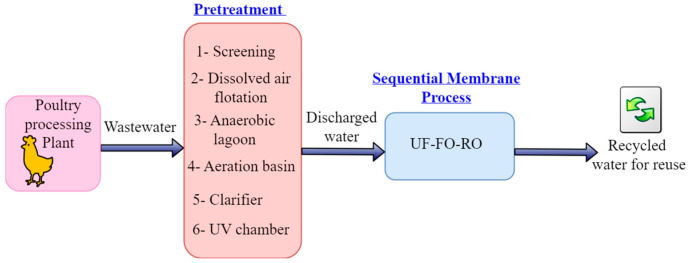
Block flow diagram of the entire process of PSWW treatment.

**Figure 2 membranes-13-00296-f002:**
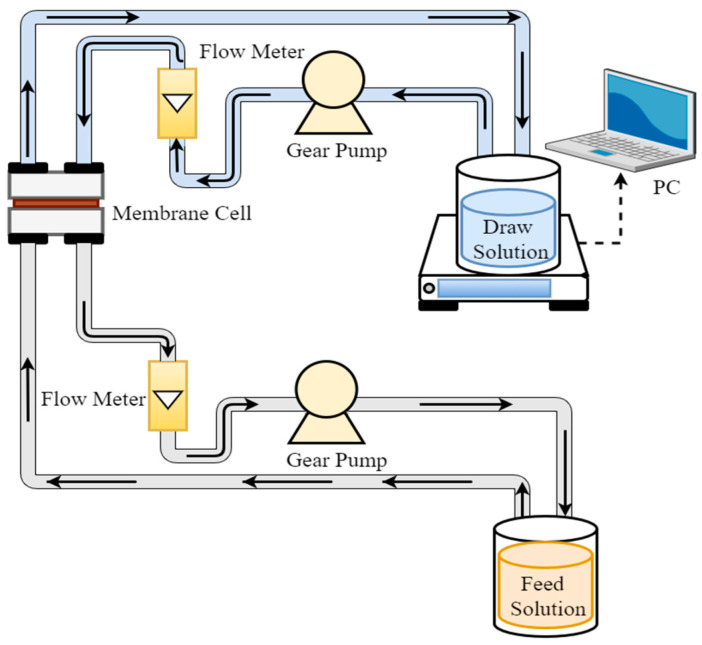
Schematic of a typical bench-scale FO system.

**Figure 3 membranes-13-00296-f003:**
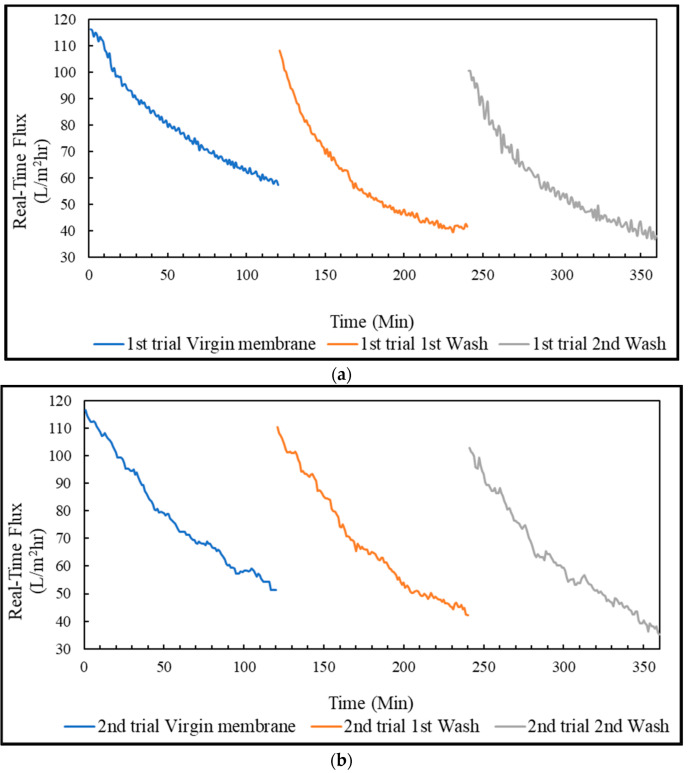
PSWW treatment with UF membrane: (**a**) 1st trial of UF membrane; (**b**) 2nd trial of UF membrane.

**Figure 4 membranes-13-00296-f004:**
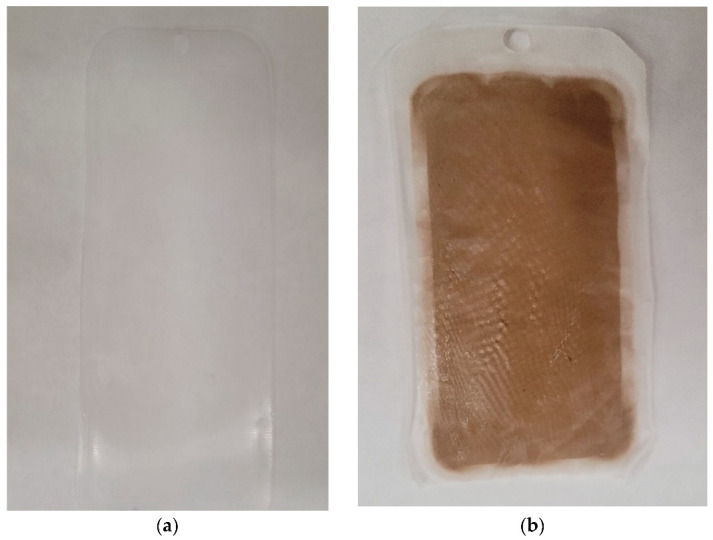
CTA FO membrane: (**a**) uncoated membrane; (**b**) coated membrane.

**Figure 5 membranes-13-00296-f005:**
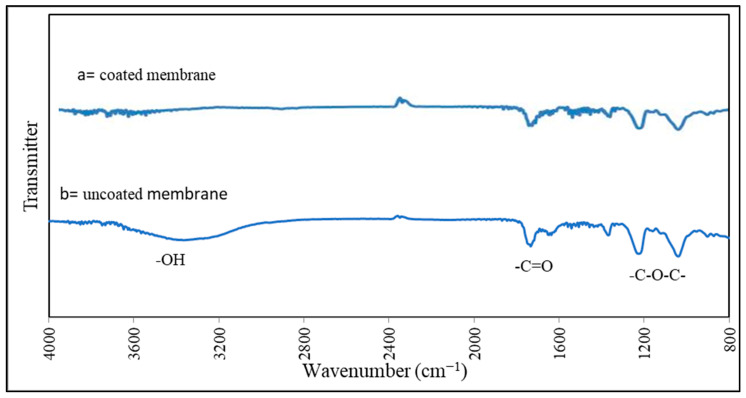
FTIR spectroscopy: (**a**) coated membrane; (**b**) uncoated membrane.

**Figure 6 membranes-13-00296-f006:**
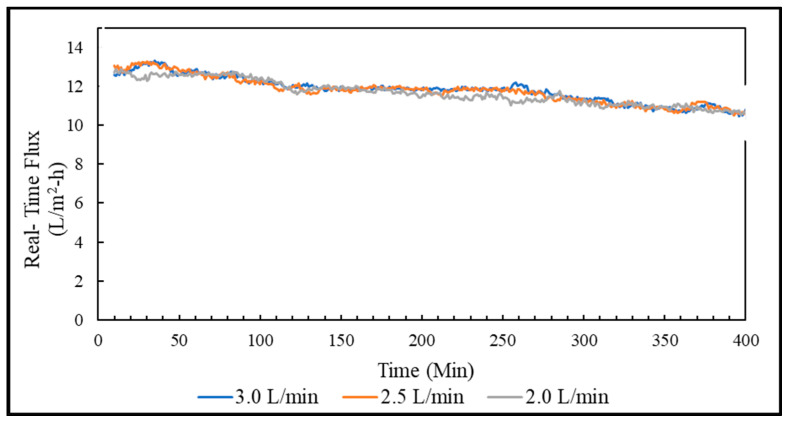
Flow rate comparison of FO system shown as moving average 10 data points.

**Figure 7 membranes-13-00296-f007:**
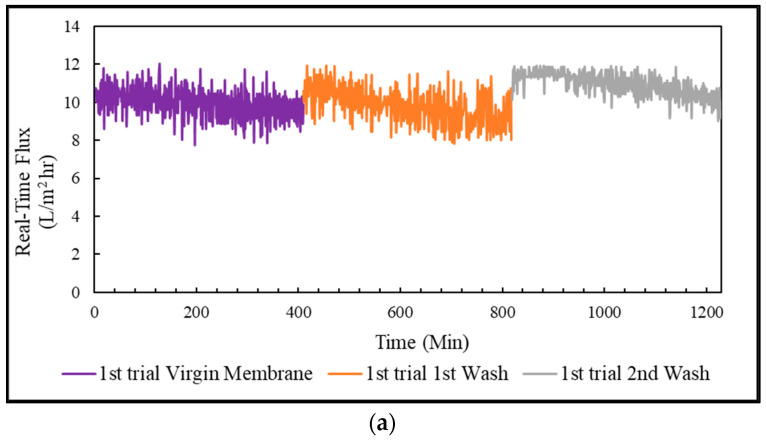
PSWW treatment in FO mode: (**a**) 1st trial of FO mode; (**b**) 2nd trial of FO mode.

**Figure 8 membranes-13-00296-f008:**
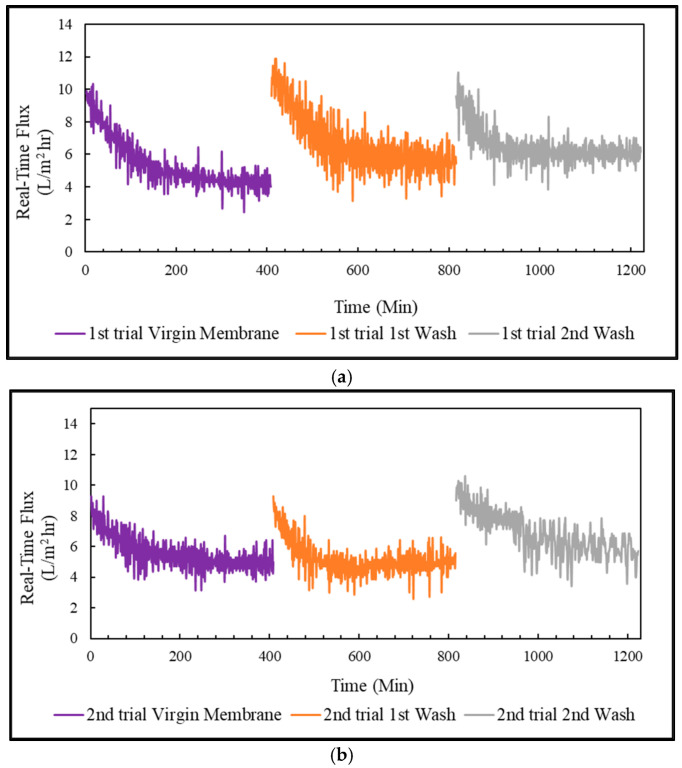
PSWW treatment in PRO mode: (**a**) 1st trial of PRO mode; (**b**) 2nd trial of PRO mode.

**Figure 9 membranes-13-00296-f009:**
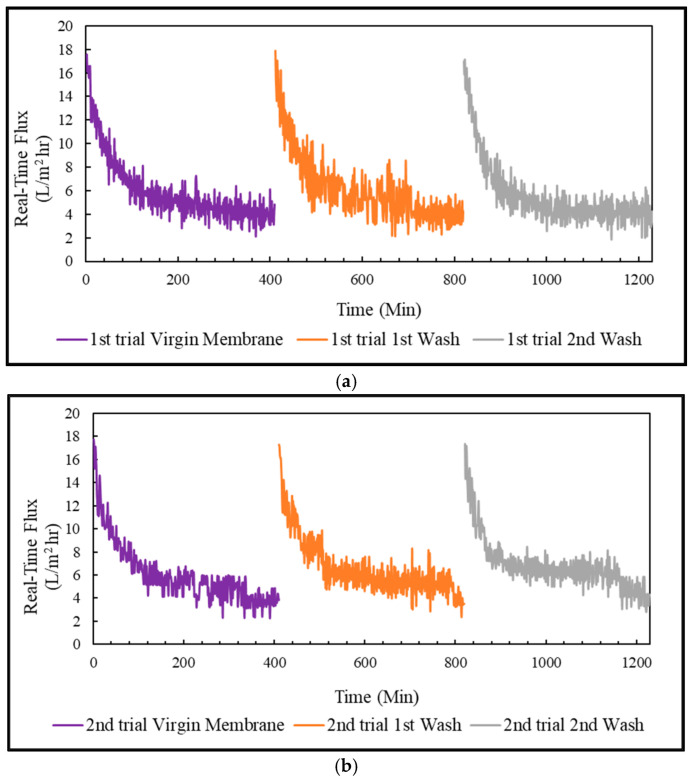
PSWW treatment with the coated membrane in PRO mode: (**a**) 1st trial of the coated membrane in PRO mode; (**b**) 2nd trial of the coated membrane in PRO mode.

**Figure 10 membranes-13-00296-f010:**
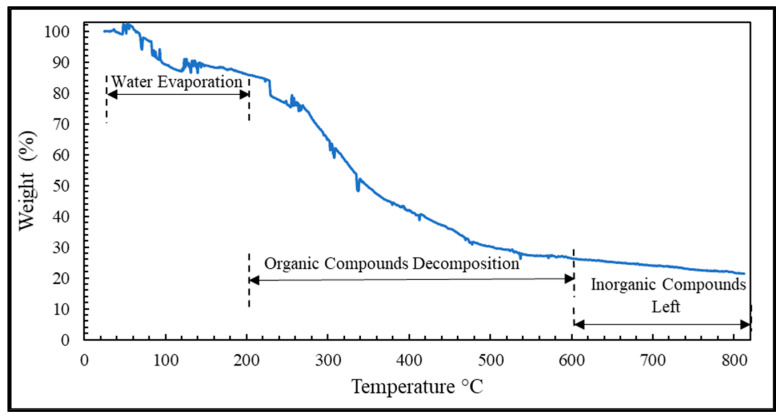
TGA of UF membrane foulants.

**Figure 11 membranes-13-00296-f011:**
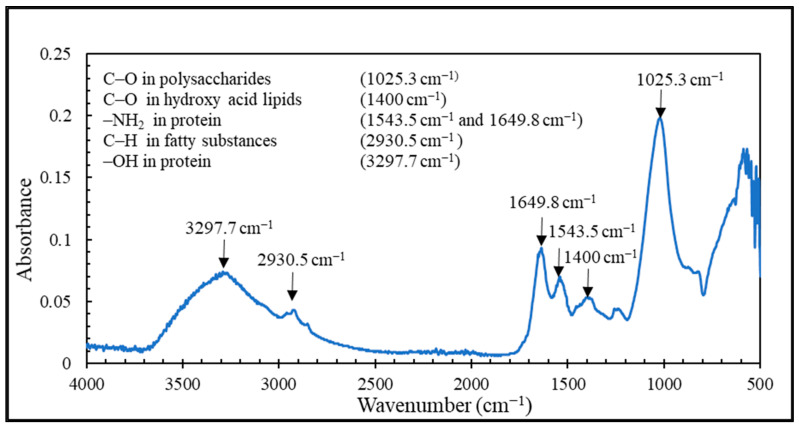
FTIR spectroscopy of UF membrane foulants.

**Table 1 membranes-13-00296-t001:** Standard Method for Examination of Water and Wastewater.

	Standard Method	Reference Method
Turbidity	180.1	LaMotte
TS	2540 B	Hach 8271
TSS	2540 D	Hach 8158
TDS	2540 C	Hach 8163
TFS	2540 E	Hach 8276
TVS	2540 E	Hach 8276
COD	5220 D	Hach 8000
TN	4500 N–E	Hach 10072
TP	4500 B–C	Hach 10127

**Table 2 membranes-13-00296-t002:** UF membrane specification [13].

Parameters	
Feed Type	Industrial
pH	1–11
Pore size/MWCO	30,000 Da
Polymer	PES
Maximum Operating Temperature	55 °C

**Table 3 membranes-13-00296-t003:** FO membrane specification [15].

Parameters	
Feed Type	Sea Water
pH	3–7
Membrane material	CTA
Maximum Operating Temperature	50 °C
Minimum Transmembrane Pressure	34 kPa
Maximum Chlorine	2 ppm

**Table 4 membranes-13-00296-t004:** RO membrane specification [16].

Parameters	
Feed Type	Industrial wastewater
pH	2–11
Membrane material	Polyamide
TDS Rejection	98.11%
Minimum Transmembrane Pressure	34 kPa
Maximum Chlorine	2 ppm

**Table 5 membranes-13-00296-t005:** PSWW characterization at the discharge stage and after UF treatment.

Parameters	Discharge Stage Water	UF Permeate	USEPA Discharging Limits [17]
Turbidity	0.6 NTU	0.0 NTU	-
COD	30 ±1.2 mg/L	19 ± 0.1 mg/L	-
TS	1830 ± 13.2 mg/L	1378 ± 20.3 mg/L	-
TSS	Not detected	Not detected	30 mg/L
TDS	1830 ± 13.2 mg/L	1378 ± 20.3 mg/L	-
TVS	366 ± 14.0 mg/L	313 ± 11.5 mg/L	-
TFS	1464 ± 8.1 mg/L	1065 ± 31.4 mg/L	-
TN	107 ± 2.0 mg/L	94 ± 1.6 mg/L	8 mg/L
TP	36 ± 1.5 mg/L	22 ± 0.3 mg/L	-

-: Not reported.

**Table 6 membranes-13-00296-t006:** Comparison of UF effectiveness with previous research.

Parameters	UF Permeate This Study	Jason et al. [18]	Marchesi et al. [19]
COD	36.7%	95%	76.7%
TS	24.7%	85%	-
TVS	14.5%	-	-
TFS	27.3%	-	-
TN	12.1%	86%	41.9%
TP	38.9%	-	-

-: Not reported.

**Table 7 membranes-13-00296-t007:** FO permeate characterization for FO mode, PRO mode, and coated membrane in PRO mode.

Parameters	FO Mode	PRO Mode	Coated Membrane in PRO Mode
COD	Not detected	Not detected	Not detected
TS	130 ± 12.3 mg/L	218 ± 11.4 mg/L	288 ± 13.8 mg/L
TSS	Not detected	Not detected	Not detected
TVS	46 ± 7.3 mg/L	52 ± 8.6 mg/L	55 ± 14.6 mg/L
TFS	84 ± 4.9 mg/L	166 ± 2.8 mg/L	233 ± 0.8 mg/L
TN	59 ± 0.5 mg/L	82 ± 1.3 mg/L	90 ± 1.6 mg/L
TP	Not detected	Not detected	Not detected

**Table 8 membranes-13-00296-t008:** Comparison of FO mode removal efficiency with previous research.

Parameters	FO Mode This Study	Ezugbe et al. [22]	Gao et al. [23]	Gao et al. [24]
COD	100%	60%	96%	96.5%
TS	90.5%	-	-	-
TVS	85.3%	-	-	-
TFS	92.1%	-	-	-
TN	37.2%	-	89%	89.4%
TP	100%	63%	100%	95.4%

-: Not reported.

**Table 9 membranes-13-00296-t009:** RO permeate characterization.

Parameters	RO Permeate mg/L	Removal Efficiency (%)	Meiramkulova et al. [26]
COD	Not detected	100%	99.6%
TS	Not detected	100%	-
TVS	Not detected	100%	-
TSS	Not detected	100%	-
TFS	Not detected	100%	-
TN	22 ± 1.7	62%	-
TP	Not detected	-	-

-: Not reported.

## Data Availability

The intermediate data in this study are available on request from the corresponding author.

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
