# Peer review of "A Sequential Membrane Process of Ultrafiltration Forward Osmosis and Reverse Osmosis for Poultry Slaughterhouse Wastewater Treatment and Reuse"

_membranes, 2023, doi:10.3390/membranes13030296_

Round 1

Reviewer 1 Report

1. I suggest that authors should provide more  detailed information  about their goals and considerations  when such an important and difficult problem is investigated. The simplified diagram shown on Figure 1  seems unsufficient to provide a solution of wastewater treatment. Each membrane method ( UF, FO,RO)  has, above efficiency, certain disadvantages and application field, also  problems to handle and dispse of concentrates, draw solutions and backwashes. The proposed wastewater (PPSW) treatment technique should account for all these problems. I suggest that a small space  in the article should be given to the discussion of these problems.

2. Discussion of the obtained experimental results should, to my mind, evaluate reasonable limits for RO recoveries, Volume concentration ratio (VCR) and backwash water amounts that can be reached in the industrial technological scheme.

3.The paragraph 3.4.4. states:" VCR was not high because the PSWW was pretreated with UF which removed  rest of the pollutants"...

It sounds strange as the goal of FO treatment should obviously be to reach a maximum VCR degree. Moreover, the lowerr VCR is, the higher is the FO process efficiency.

4. The statement in paragraph 3.4.6 sounds also surprizing:"There are two types of fouling based on cleanibility: reversible fouling and irreversible fouling". It seems to me that authors should support their description by description of nature of foulants, such as colloidal, particulate, sparingly soluble, colloidal etc.

Author Response

All the authors thank you for the helpful comments to improve the quality of our manuscript in the first-round review. Please check our point-by-point response in the attached file.

Reviewer 2 Report

Please find attached

Author Response

(The authors gave the same response as above.)

Reviewer 3 Report

This is a work about exploring a combination of ultrafiltration, forward osmosis and reverse osmosis for the deep treatment of poultry slaughterhouse wastewater. some specific suggestions are as follows:

1.    The title is not clear enough, the feed water of this work has been treated using as mentioned in line77-79, so there is not appropriate for “Poultry Slaughterhouse Wastewater” in the title.

2.    There is a writing error in Table 6, with % more written in front of the references.

3.    In Figure 3, are these three parallel experiments? If so, it is better to draw them on one figure with error bars.

4.    The membrane fouling after surface modification is more serious. Can the author provide more powerful reference support or characterization for this phenomenon in line 341-346?

5.    There is a writing error in line 257, “11.7 L/m2-hr” should be “11.7 L/( m2·h)”.

6.    Whether continuous flow experiments have been conducted to monitor the operation effect and ultrafiltration membrane fouling under long-term operation conditions?

7.    In line 99-104, how does the cleaning parameters of ultrafiltration membrane determined? the cleaning intensity would affect the stability of the membrane?

8.    For the modified FO membrane, is there any characterization of microscopic morphology to prove that the pore size of modified FO membrane was smaller, such as SEM, etc.

9.    Provide more detailed steps on the configuration of L-DOPA solution in section 2.3?

10. Provide the specific steps and methods modification of FO membrane in line118-120?

11. The specific type of instrument should be provided in line 124-127.

12. What is the volume of feed and draw solution for FO system, and what is the membrane area in 2.4?

13. In the mode of pressure-retarded osmosis,what is the pressure applied in the system?

14. The full name of pollutants should be written when they first appear, such as COD,TSS, etc.

15. From the perspective of raw water quality, the concentration of organic matter in pollutants is not high (COD 30±1.2 mg/L), mainly inorganic substances. In this case, the combination of UF, FO and RO membrane processes is used. Will the cost be raised, or can the effluent quality be guaranteed only by the combination of UF and RO processes?

16. L-DOPA was coated on the membrane by the FO system. Was the stability of this modification examined during the experiment? How to consider the stability and morphology of the film before and after use.

17. The concentration of TN in effluent after RO is still very high (22± 1.7mg /L) at Table 9, which obviously does not meet the requirements of water reuse. In view of this deficiency, can the author propose further improvement measures to reduce the concentration of TN?

18. Conclusion is similar with the abstract, a clearer and more concise conclusion would be more appropriate.

Author Response

(The authors gave the same response as above.)

Reviewer 4 Report

The quality of the paper should be enhanced. Major comments:

1. Fig.1 is very shallow and meaningless. Please replace it with the flow diagram of full poultry processing and highlight UF-FO-RO in it.

2. Table 1 is also meaningless, as it has been specified in the text. Please replace it with the used instruments, reagents and units for each testing indicators.

3. Why did you coat the FO membrane instead of using a commercial one?

4. Fig.2 is not beautiful and needs major improvement.

5. Why is there no report of COD, TS, TN, TP in USEPA Discharging Limits? They are common indicators of water quality.

6. Line 181-187 & Table 6: What is the basis for the comparison of removal rate? The applied condition of UF is different in other studies.

7. The drawing of Fig.3 seems to have big problems. 1) the data of the 3 parallel trials should be drawn as a curve as the average value, not in 3 separate figures. 2) the data after washing should not be drawn from 0 min, but 120 min. The authors should refer to the references with regard to membrane cleaning and see how they present the flux curve. For example:

https://doi.org/10.1016/j.memsci.2022.121013

8. Fig.4: Why the coated membrane looks like suffering membrane fouling? Would it suffer more severe fouling in the application?

9. Fig.5: The wavenumber should be lower in the left to the higher in the right?

10. Line 258: It seems that a higher flow rate is always better for improving water flux. Can it be higher than 3 L/min? Is there an upper limit and what is the restriction factor? The authors should think and discuss more deeply. The following references may contribute to the discussion.

https://doi.org/10.1016/j.desal.2021.115447

https://doi.org/10.1016/j.watres.2021.117146

11. The discussion of membrane fouling: What is the main foulant on FO and UF membrane?

12: “Section 4.1”? Is there Section 4.2, 4.3…?

Author Response

(The authors gave the same response as above.)

Round 2

Reviewer 2 Report

The authors responded the most of my comments satisfactorily. The manuscript could be published. 

Author Response

Thank you for the helpful comments.

One native English speaker, a senior program coordinator working with us, proofread our manuscript, and she was satisfied with our written English.

We added one reference related to our discussion in Subsection 3.4.1.

Reviewer 4 Report

The authors have fixed most of the concerns, but there are some not well addressed.

1. The reply to Comment 10: "a higher flow rate is always better for improving water flux and less membrane fouling" is not true; moreover, it may cause some other problems. Make a discussion with the suggested references.

2. Fig. 6-9: the lines are crossed and not clear at all. Make some changes to the presentation styles.

3. Can not understand why UF membrane foulants were composed of 20% water, 60 % organic, and 20 % inorganic compounds according to Fig. 10. Please add the key points and stages in the figure to show the information.

4. Fig. 11: How can you know the peaks are from the foulants, not influenced by the membrane? Besides, the information of the functional group should be noted in the figure.

5. The font and size of the characters in all the figures are not uniform throughout the whole paper. Check it.

Author Response

The authors have fixed most of the concerns, but there are some not well addressed.

1. The reply to Comment 10: "a higher flow rate is always better for improving water flux and less membrane fouling" is not true; moreover, it may cause some other problems. Make a discussion with the suggested references.

Response: We took your sugestions and fixed the issue. Please check  the revision in Section 3.4.1 “We need to point out that the water flux does not always increase with the increase of the crossflow rate of feeding water because high crossflow rate can cause some issues, e.g., feed channel pressure drop [21]”.

2. Fig. 6-9: the lines are crossed and not clear at all. Make some changes to the presentation styles.

Response: Fixed. Please check the revised figures 6-9.

3. Can not understand why UF membrane foulants were composed of 20% water, 60 % organic, and 20 % inorganic compounds according to Fig. 10. Please add the key points and stages in the figure to show the information.

Response: Fixed. Please recheck Figure 10. The composition of membrane foulant on the membrane surface was examined with TGA. The rough composition percentage of moisture, organic and inorganic matter can be deduced from the dynamic TGA curve along with the increasing temperature.

4. Fig. 11: How can you know the peaks are from the foulants, not influenced by the membrane? Besides, the information of the functional group should be noted in the figure.

Response: Fixed. Please recheck Figure 11 and section 2.5 for updated information “For the FTIR spectroscopy, the foulants were collected from the membrane surface, air-dried at room temperature, and then tested for detecting functional groups. The weight percentage of moisture, organic and inorganic compounds present in the foulants was determined by thermogravimetric analysis (TGA). The TGA was conducted starting from room temperature, increasing by 10 °C /min and reaching up to 800 °C “

5. The font and size of the characters in all the figures are not uniform throughout the whole paper. Check it.

Response: Thanks for the comments, and we adjust the fonts and their size in the figures. Please check the revised Figures 2, 3, and 5-11.

In addition, one native English speaker, a senior program coordinator working with us, proofread our manuscript, and she was satisfied with our written English.  

Round 3

Reviewer 4 Report

No further comments.